# Lessons Learned from Translating Genome Sequencing to Clinical Routine: Understanding the Accuracy of a Diagnostic Pipeline

**DOI:** 10.3390/genes15010136

**Published:** 2024-01-22

**Authors:** Joohyun Park, Marc Sturm, Olga Seibel-Kelemen, Stephan Ossowski, Tobias B. Haack

**Affiliations:** 1Institute of Medical Genetics and Applied Genomics, University of Tübingen, 72076 Tübingen, Germany; joohyun.park@med.uni-tuebingen.de (J.P.); olga.kelemen@med.uni-tuebingen.de (O.S.-K.); stephan.ossowski@med.uni-tuebingen.de (S.O.); 2Center for Rare Diseases, University of Tübingen, 72076 Tübingen, Germany

**Keywords:** genetic diagnostics, genome sequencing, molecular diagnostics

## Abstract

The potential of genome sequencing (GS), which allows detection of almost all types of genetic variation across nearly the entire genome of an individual, greatly expands the possibility for diagnosing genetic disorders. The opportunities provided with this single test are enticing to researchers and clinicians worldwide for human genetic research as well as clinical application. Multiple studies have highlighted the advantages of GS for genetic variant discovery, emphasizing its added value for routine clinical use. We have implemented GS as first-line genetic testing for patients with rare diseases. Here, we report on our experiences in establishing GS as a reliable diagnostic method for almost all types of genetic disorders, from validating diagnostic accuracy of sequencing pipelines to clinical implementation in routine practice.

## 1. Introduction

Exome sequencing (ES) has revolutionized disease gene identification, drastically improving and accelerating clinical diagnosis [1]. It is a highly efficient diagnostic method, because it essentially focuses on the protein-coding regions that comprise approximately 1.5% of the genome. Despite the enormous progress in human genetic research, ES can only solve between 5% and 50% of cases with suspected genetic disorders, varying greatly between different disease groups [2,3,4]. The inability to find the genetic causes can have many reasons such as multigenic inheritance, reduced penetrance, variable expressivity, or non-coding variants. Genome sequencing (GS) is the obvious next step to close this gap as it enables detection of almost all forms of genomic variation, including single nucleotide variants (SNVs), small insertion/deletion variants (InDels), copy number variants (CNVs), and repeat expansions, as well as complex structural variants (SVs) such as deletions, duplications, inversions, translocations and mobile element insertions (MEIs), with a single test [5]. Thus, GS can detect additional potentially disease-causing variants, especially those affecting non-coding regions, substantially increasing the diagnostic yield for patients with rare disorders [6,7,8].

Now that GS has become more accessible and affordable, it is widely used for human genetic analyses such as population database compilation, genome-wide association studies, rare disease research, and clinical care, leading us one step closer to defining and better understanding the full spectrum of human genetic variation. Although GS is now commonly used and well investigated, its implementation in clinical routine can be challenging. The amount of data that needs to be produced, analyzed, interpreted and reported in a certain timeframe can be overwhelming. In particular, the number of variants of unknown significance increases significantly, and variants in regions that are difficult to map can be technically difficult to interpret or even missed. Although bioinformatic tools are available for the detection of different genomic variations, their performance has to be evaluated before they are integrated into clinical routine testing, where they can replace conventional approaches.

Here, we report our experience of moving from short-read ES to short-read GS on a routine basis and the benefits of GS in terms of variant calling and diagnostic yield. We introduce our quality control schemes and technical validation strategies using Genome in a Bottle (GiaB) benchmarks, which we applied to assess the sensitivity and specificity of detected genetic variations. Furthermore, we evaluated the number of rare variants identified in our diagnostic ES and GS pipeline as well as our diagnostic report outcomes of 2022. Moreover, we share our state-of-the-art workflow for efficient variant prioritization and interpretation. These efforts may be essential to assure diagnostic accuracy and avoid missed diagnoses, and arebeneficial for variant interpretation.

## 2. Materials and Methods

### 2.1. Diagnostic Exome and Genome Sequencing

For comparison of ES and GS, we selected nearly 3000 index cases sequenced at the Institute of Medical Genetics and Applied Genomics Tübingen (Tübingen, Germany) in 2022. The dataset consisted of 1016 ES and 1977 GS samples. Trio-analysis was performed on 22.3% of ES samples and 1.8% of GS samples. The median age of patients who underwent ES and GS was 7 (interquartile ranges Q1–Q3: 4–19) and 49 (Q1–Q3: 29–61) years, respectively (Appendix A). The samples (peripheral blood) were sent for routine diagnostic GS from different external and internal physicians. For an unbiased evaluation, we included all samples regardless of the phenotype, disease group and age of disease onset. This included all samples from patients with high and even low suspicion of genetic disease in whom a genetic disorder is listed as a possible differential diagnosis requiring a genetic workup, with or without a positive family history. The disease group distribution is listed in Appendix A. The majority of samples were of European descent (85.23% in ES and 93.27% in GS). An overview of all ancestries can be found in Appendix A.

The ES samples were processed with the SureSelect Human All Exon v7 kit (Agilent, Santa Clara, CA, USA) and sequenced using the NovaSeq6000 system (Illumina, San Diego, CA, USA) as 100 bp paired-end reads. On the 49.25 Mb target region, an average depth of 150× was the aim for ES (Table 1).

For GS, we generated sequencing libraries using the TruSeq DNA PCR-Free kit (Illumina, San Diego, CA, USA). Sequencing was conducted on a NovaSeq 6000 system (Illumina, San Diego, CA, USA) as 150 bp paired-end reads with an average target depth of 38×.

### 2.2. Data Analysis Pipeline and Decision Support System

All the data presented in this work were processed with our data analysis pipeline megSAP. megSAP is freely available at https://github.com/imgag/megSAP. A list of the used open-source tools and databases can be found at https://github.com/imgag/megSAP/blob/master/doc/dna_single_sample.md. GRCh38 with the false duplications masked [9] was used as reference genome for mapping. After mapping, small variant calling, CNV calling and SV calling were performed. CNV calling is based on the depth of coverage in an exon for ES and 1000 bp bins for GS. The ExpansionHunter tool was applied to screen for 39 different repeat motives/regions (see Appendix A) [10].

The following databases and prediction tools were used for annotation of variants: gnomAD [11] for population allele frequency; SpliceAI [12] and MaxEntScan [13] for splicing prediction; ClinVar [14] and HGMD^®^ [15] for known pathogenic variants; CADD [16], REVEL [17], and AlphaMissense [18] as pathogenicity predictors.

After data analysis using the megSAP pipeline, the QC metrics and variants of each sample were imported into an in-house NGS database (NGSD). The QC metrics were stored in the database to allow for visualization of the sequencing metrics, automated outlier detection and monitoring of trends over time. Storing all variants in a database came with many advantages. For example, they were used to detect pipeline-specific artefacts, i.e., variants that are frequently called for in the pipeline but are not listed in population databases like gnomAD. Another typical use-case of the variant database is to evaluate the clinical phenotype of all samples in which a specific variant is found. This enables rapid detection of recurring disease-causing variants.

Besides the analysis pipeline megSAP and the database NGSD, there is a third important component used in our diagnostics system: the decision support system GSvar. This displays all variants of a sample, including annotations, as a table and allows filtering of the variants according to pre-defined or custom filters. GSvar also supports the classification of variants by showing annotations from several databases and prediction tools. All information that is manually added to a variant, e.g., comments and classification, is stored in the NGSD. More information about GSvar can be found at https://github.com/imgag/ngs-bits/blob/master/doc/GSvar/index.md.

### 2.3. Pipeline Validation

When using an NGS data analysis pipeline for diagnostics, a thorough benchmarking of its performance is required. We benchmarked the small variant calling performance of megSAP based on the GiaB reference sample NA12878 using the reference variant list and high-confidence region v4.2.1.

### 2.4. Quality Control

Quality control (QC) for NGS data was performed at different levels (raw sequencing data, mapped reads and variants). Only the most important QC metrics are discussed here. A full list of the calculated quality metrics can be found in Appendix A. Raw data QC was performed on the FASTQ files after sample demultiplexing. Here, the most important metrics are the read count, number of bases sequenced and percentage of bases with a minimum quality score of Q30. Mapping QC metrics were calculated from the BAM/CRAM file after read mapping. The most important QC metrics for mapping are the percentage of reads mapped to the reference genome, the average insert size, percentage of reads marked as duplicates and the average read depth in the target region. To estimate the uniformity of coverage, the percentage of the target region covered at least 20× was used. To detect contamination, we used the metric “SNV allele frequency deviation”, which captures the change in heterozygous variant allele frequency caused by contaminations with other human samples. For the variant-level QC, the following QC metrics were used: variant count, percentage of variants listed in the dbSNP database, percentage of homozygous variants and percentage of InDel variants.

Another important aspect of QC is to detect sample swaps. For single samples, we could only check the gender. For family analyses, e.g., trios or multiple affected/unaffected family members, the percentage of shared variants and the correlation of the genotypes was used to verify sample relationships. All QC steps described here were performed using tools from ngs-bits, which is freely available at https://github.com/imgag/ngs-bits.

### 2.5. Coverage Statistics

For the comparison of coverage in diagnostically relevant regions, we evaluated the coverage of 4739 OMIM (Online Mendelian Inheritance in Man; omim.org) genes on autosomes and chromosome X. We included exons and 20 bp flanking splice regions, which totals to a region of 14.01 Mb. To have a uniform cohort for chromosome X, we excluded male samples from this evaluation. In addition, we assessed the average number of repeat motifs/regions that were flagged as LowDepth by the ExpansionHunter across the ES dataset [10].

### 2.6. Variant Count Statistics

To compare the number of variants assessed in the diagnostic process, we applied the following filters: For rare small variants, we filtered for gnomAD MAF ≤ 0.1% and an NGSD count ≤ 10. For private variants, we filtered for gnomAD MAF = 0% and an NGSD count ≤ 1. As the criterion NGSD count ≤ 1 is very sensitive to relatives of a sample contained in the database, we excluded all samples with relatives, leaving 620 European and 105 non-European samples for ES and 1711 European and 114 non-European samples for GS. The number of CNVs was reduced effectively by filtering for high-quality calls involving at least 2 regions (exons/bins), an in-house allele frequency ≤ 5%, and an overlap with an OMIM gene. For SVs, we counted deletions, duplications, inversions and insertions of good quality (quality ≥ 100, supported by at least 5 paired-end reads, no filter entries) with an in-house allele frequency ≤ 1% and overlap with an OMIM gene.

### 2.7. Diagnostic Filtering Strategies

For data filtering the following criteria were used: variant quality metrics, variant frequency in gnomAD, variant count in NGSD, phenotype-based target regions, pathomechanism, and mode of inheritance. A simplified illustration of our standard filtering scheme is shown in Figure 1. For the initial stringent prioritization, small variants were filtered for rare and coding variants for each possible mode of inheritance. Rare variants were defined by MAF < 0.1% in gnomAD and an NGSD count ≤ 5 for dominant and MAF < 1% in gnomAD and an NGSD count ≤ 40 for recessive models. Known pathogenic intronic variants according to ClinVar and NGSD were highlighted and not filtered out throughout the data interpretation process. Intronic and splice region variants with predicted splice effects, e.g., SpliceAI score > 0.5, were also kept during filtering. An extended search included evaluation of non-coding regulatory variants (especially in the presence of a characteristic phenotype for a particular genetic disorder), CNVs/SVs as a second hit when a single pathogenic heterozygous variant was detected, and mosaic variants. Additional filtering options (e.g., variant type, variant impact (high/moderate/low), and target region based on HPO phenotype [19]) were used for variant prioritization on a per-sample basis. CNVs and SVs were filtered by size, variant type, NGSD allele frequency and gene content. To enhance the reliability of the detected SVs, we used quality-based filtering, which included a minimum number of supporting paired-end reads or split-reads, as well as removing SVs located in decoy sequences or in regions with a high density of SV break points. Trio-analysis uses different filtering strategies targeting (i) rare, de novo coding, regulatory and structural variants; (ii) homozygous and compound-homozygous variants in known recessive disease genes; (iii) loss of heterozygosity (LOH); and (iv) rare variants in X-chromosomal and mitochondrial genes. To avoid potential pitfalls, special attention is devoted to (i) germline mosaicism, (ii) inherited variants in imprinting genes/regions, (iii) variants in genes that induce clonal hematopoiesis of indeterminate potential (CHIP), and (iv) genes that underlie variable penetrance [20].

### 2.8. Variant Interpretation

Single small variants, CNVs and SVs were classified according to the classical guidelines of the American College of Medical Genetics and Genomics (ACMG) and the Association for Molecular Pathology [21,22]. We determined the number of solved cases across all disease groups. A case was considered solved when a pathogenic (class 5) or likely pathogenic (class 4) variant was identified matching the phenotype. All causal variants were uploaded to Clinvar. A case was considered to have a possible causal finding when a variant of uncertain significance (VUS, class 3) matching the phenotype was detected and reported. We also systematically screened for solved cases with causal variants that are non-coding variants, short-tandem repeat expansions (STRs), complex SVs, or smaller CNVs (<2 exons) not expected to be detected by ES.

## 3. Results

### 3.1. Uniformity of Coverage and Gaps

For diagnostic sensitivity, it is very important to have as few coverage gaps as possible. Thus, we compared the gaps defined as regions with less than 20× coverage between GS and ES. In the ES cohort (*n* = 412), an average coverage of 157× was achieved, but still 2.80% of the OMIM region had insufficient coverage of less than 20×. In GS (*n* = 1178), the average depth was 39×, but only 0.23% of the OMIM region was covered below 20×. This evaluation includes reads with a mapping quality zero. When excluding reads with a mapping quality zero, 3.26% and 0.63% low-coverage regions were present in ES and GS, respectively (Table 1). Furthermore, an average of 16 out of 39 different repeat motifs/regions (Appendix A) were flagged as LowDepth in the ES dataset.

### 3.2. Small Variant Calling Benchmarks

To determine the diagnostic performance of ES and GS, we performed our standard benchmarks on the GiaB NA12878 sample with 142× coverage for ES and 40× coverage for GS. The SNVs F1 scores were 0.9875 and 0.9955, and the InDel F1 scores were 0.9427 and 0.9858 for ES and GS, respectively. However, these benchmarks are not comparable as they are based on different target regions (exome vs. genome) and only variants in regions with at least 15× coverage were considered. We have already shown that genomes have better uniformity of coverage then exomes. Therefore, under real-world conditions, the difference in performance between genomes and exomes should be higher. To further compare the real-world diagnostic performance, we performed an additional benchmark on the protein-coding exons plus 20 bp splice regions without a minimum coverage cutoff. In this second benchmark, the SNVs F1 scores were 0.9795 and 0.9916, and the InDel F1 scores were 0.9309 and 0.9878 for ES and GS, respectively. Here, it becomes even more obvious that the performance of GS is superior to ES, especially for InDels, but also for SNVs. The sensitivity and positive predictive value (PPV) from which the F1 scores were calculated are listed in Table 2. More information about the performance of megSAP can be found at https://github.com/imgag/megSAP/blob/master/doc/performance.md.

### 3.3. Number of Variants

We identified approximately 302 ± 86 rare and 73 ± 28 private variants in an exome dataset (Figure 2, Table 3). This number depended on the ancestry of the individual and the population groups represented in the control databases. In comparison, 24,809 ± 5655 rare and 5533 ± 1746 private small variants were identified in a genome dataset, which includes non-coding areas. GS therefore results in 82-fold more rare variants and 75-fold more private variants than ES. When looking at coding and splicing regions only, an ES dataset contained 252 ± 71 rare and 61 ± 23 private variants. Similarly, a GS dataset contained 260 ± 54 rare and 61 ± 20 private variants. For GS, we found 4 ± 6 rare, high-quality CNVs overlapping with OMIM genes and 9 ± 4 rare, high-quality SVs overlapping with OMIM genes.

All variant counts above are based on samples with European ancestry. Table 3 and Figure 2 also show variant counts for non-European samples. When filtering for rare variants, the variant counts for non-European samples are generally about two times higher than for European samples. This was expected, as non-Europeans are underrepresented both in gnomAD and our in-house database NGSD. Interestingly, we found a distinct fraction of samples within the European cohort with a higher number of variants (more than three standard deviations above the mean). We checked the patient names of these outliers and found that they were mainly of Middle Eastern (predominantly Turkish) descent.

### 3.4. Diagnostic Outcomes in ES and GS

A trio-analysis was performed on 22.3% of the 1016 ES samples. In total, 25.59% of ES samples received a definite diagnosis with a pathogenic or a likely pathogenic variant, and 27.46% had at least one VUS in a gene associated with the patient’s presenting clinical symptoms (Figure 3, Appendix A). Two pathogenic STRs were detected using ExpansionHunter in ES, which were subsequently validated by conventional PCR-based methods. For GS, 416 of the 1977 index patients (21.04%) had a definite diagnosis. A total of 514 patients had at least one VUS reported matching the phenotype, representing 26% of our GS reports. Most causal variants identified in GS (*n* = 1977) were either small variants in coding regions, splicing variants (+/*−* 20 bp) or large CNVs, which are usually expected to be covered by ES. However, in 37 (8.89%) out of the 416 solved diagnostic cases, other causal variations were reported which might not have been detectable with ES. These included 13 variants in non-coding intronic regions, 12 repeat expansions, 10 small copy-number variations (≤3 exons) and 2 structural variants, all of which were classified as likely pathogenic or pathogenic and considered clinically relevant.

## 4. Discussion

Here, we report on our practical experiences and analysis strategies acquired during the establishment of GS as a first-line diagnostic test and prospectively evaluated in routine clinical care. We compared our real-life ES and GS data generated in 2022. Our diagnostic benchmarks have proven that GS has superior performance in variant calling for all types of variations (Table 2).

Uniformity of coverage is also a very important factor in diagnostics, as low-coverage regions, also referred to as diagnostic gaps, lead to lower sensitivity in detecting disease-causing variants. Our benchmarking study showed that in coding regions of OMIM genes in the GS datasets, there were 5-fold less low-coverage regions (<20×) than in ES. The uneven coverage in ES is mainly caused by biases in enrichment probe binding and PCR amplification, which affect regions with extreme GC content. These biases have been known to create coverage gaps in clinically relevant genes as well as low-quality and false-positive CNV calls [23]. Thus, with the introduction of PCR-free library preparation for GS, we have observed a drastic improvement in sequencing quality in terms of PCR duplicates, PCR artefacts and uniformity of coverage. Masking of false duplications in the GRCh38 reference genome is also crucial to make the affected genes accessible.

In our European GS cohort, approximately 260 ± 54 rare and 61 ± 20 private small variants were within coding regions, while 24,809 ± 5655 rare and 5533 ± 1746 private small variants were found in the entire genome. The previous gnomAD analyses identified approximately 200 rare, coding variants (gnomAD MAF < 0.1%) and a mean of 27 ± 13 private variants in a new exome dataset [24]. Our rare variant count was higher possibly because of two reasons: (1) our analysis pipeline was tuned for sensitivity, allowing for the detection of more false-positives, while the gnomAD project applied stringent quality filtering, both on variant and genomic region levels, to avoid false-positive variants; and (2) our European cohort contained about 15% Middle Eastern descent samples, which significantly increased the mean and standard deviation. When comparing the rare/private variant counts of GS and ES in the coding and splicing regions, there was no significant difference in the absolute numbers. Nevertheless, a rigorous variant prioritization strategy is needed to cope with the large number of rare non-coding variants in GS, and we presented our filtering strategies in this paper (see Methods, Figure 1).

Due to the challenge of covering the sequencing costs for diagnostic trio-GS and because trio-analysis is recommended for early childhood diseases, trio-ES was preferably conducted for pediatric and prenatal cases in 2022. Consequently, index patients who underwent ES in 2022 had a significantly younger median age (ES: 7 years, GS: 49 years), and thus the utilization of diagnostic trio-analysis was considerably higher in ES (ES: 22.9%; GS: 1.8%). Younger age at disease onset and the use of a trio-approach are both factors contributing to a higher probability of establishing a firm diagnosis, as demonstrated by numerous previous studies [25,26,27]. This possibly explains the slightly higher overall diagnostic yield in ES for 2022 (25.59%, Appendix A). Several recent studies have delved into the cost-effectiveness of trio-GS in pediatric cases, signaling for the necessity to extend diagnostic options [28,29].

In ES, two cases were solved using STRs, whereas 12 were solved in GS. Since many pathogenic repeat motifs lie in intronic or untranslated regions, these regions may be underrepresented or not covered depending on the exome kits. In our exome datasets, on average 16 of the 39 repeat regions/motifs were flagged as LowDepth by the ExpansionHunter tool. This indicates that GS has a higher sensitivity for repeat expansion disorders. However, a direct comparison of the diagnostic yield of repeat expansion detection between the two methods was not feasible in this study, as the phenotypes commonly associated with repeat disorders were not evenly represented in the ES and GS cohorts.

Overall, our data suggest an increase in diagnostic yield of almost 9% for GS compared to ES, which we estimated by the number of reported causal structural variations and variants in non-coding regions that are usually not technically covered in ES. We hypothesize that the true diagnostic utility of GS is even higher due to its higher sensitivity for all types of genomic variations and its better coverage. For a direct comparison, an ES and GS would have been needed for all cases. A recent study by Wojcik et al. demonstrated that 28% of causal variants in their cohort could not to be detected by ES [30]. However, these also included VUS which were considered as solved/likely solved, which we evaluated separately. A total of 55 (2.78%) of our cases had undergone inconclusive ES prior to GS, meaning that previously solved cases by other genetic testing were not included in this study. Furthermore, we did not pre-stratify our cases for phenotypes or factors with a high likelihood for a genetic cause, but we kept all cases with a variety of clinical conditions that had been referred for GS from different clinicians to illustrate an unbiased result of a single genetic center. Therefore, the diagnostic yield for an individual disease group should not be regarded as the overall diagnostic yield for each test. Our results rather reflect the unambiguous outcomes after diagnostic GS in 2022. Above all, more than 514 patients (26%) received a VUS as a possible cause (Appendix A). Further diagnostic work-up, such as segregation analysis, biochemical analysis, functional evaluation, specific phenotyping or other omics approaches (e.g., transcriptomics, proteomics), has been recommended for each of the reported VUSs, which will and might have already set a definite diagnosis in a number of these cases as a follow-up of the GS diagnostic report.

In a diagnostic setting, comprehensive QC is crucial to prevent potential downstream diagnostic shortcomings. We presented our QC scheme as well (see Methods, Appendix A). Monitoring the QC metrics over time allows for automated outlier detection and detection of trends over time. It is also very helpful when evaluating the benefits and shortcomings of new approaches, e.g., when transitioning from diagnostic ES to GS. From our experiences, setting up a new sequencing technology can be challenging and requires experienced personnel from different fields, involving technicians, molecular biologists, bioinformaticians and clinical geneticists. Before transferring GS to clinical practice, the tests should be validated on different levels, accredited for clinical use and monitored using appropriate quality parameters. Each laboratory should decide whether GS is feasible for routine diagnostic testing in their environment, considering the cost-effectiveness, infrastructure and personnel. When implemented, GS provides a significant advantage as it can be used to detect most genomic alterations in a single test with better quality. This may potentially even save more time and the cost spent per patient compared to using multiple tests for detecting different variant types.

The achievement of a 100% diagnostic sensitivity is still a long way off, possibly due to remaining difficulties in detecting variants, optimizing filtering techniques, and appropriately interpreting the detected variants. Some of these issues may be tackled by using complementary RNA-seq or other omics approaches [6,31,32,33,34]. In addition, the increase in read lengths using long-read (LR) sequencing technologies is expected to further advance our ability to detect complex structural variation and better characterize, e.g., repeat expansion. Of note, LR sequencing also offers the opportunity to address the current blind spot of short-read sequencing technologies, namely the detection of epigenetic modification, rendering LR-GS a putative all-in-one diagnostic approach. While first initiatives of clinical implementation of LR-GS are underway, the lack of cohorts with streamlined processed datasets linked to standardized phenotypic information remains a challenge to establishing firm variant–/gene–phenotype associations, e.g., etiological discovery gene-burden analyses. In this context, there is a broad consensus regarding the imperative for healthcare standards and infrastructures tailored to rare disorders (RDs). The efficacy of a national sequencing program and a dedicated research environment has been demonstrably robust, as evidenced by existing national frameworks [4,35,36]. These (inter)national initiatives are effectively charting the course for a comprehensive exploration of structural variants; rare variations within non-coding genes; regulatory elements of the genome; and alternative models of inheritance, including polygenic effects and instances of reduced penetrance. Within this context, there is a prevailing support that genetic data acquired with informed consent as part of routine diagnostic procedures should be made accessible to the research community. This collaborative sharing is intended to propel the discovery of novel disorders and provide valuable insights for the clinical reporting process. However, the operationalization of these principles in routine clinical settings poses significant challenges and remains largely in the formative stages at a national level. However, at the level of a tertiary clinical care center, we have shown with the introduction of GS that the implementation of such concepts is feasible, and we will continue to strive to achieve maximum diagnostic yield for the benefit of patients.

## Figures and Tables

**Figure 1 genes-15-00136-f001:**
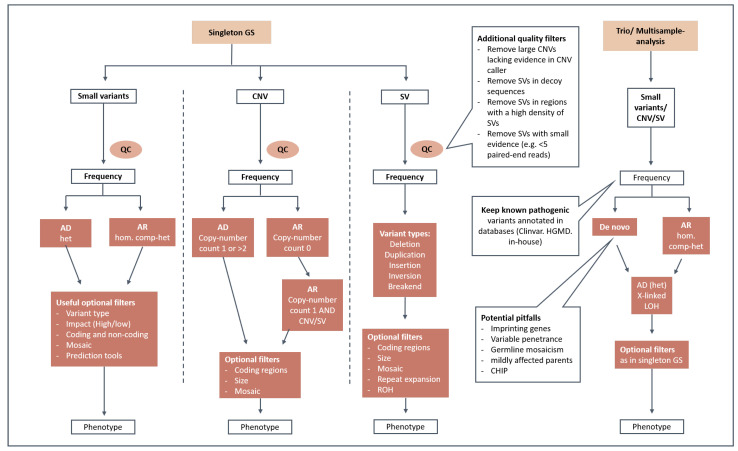
Prioritization strategy/standard filtering scheme. This graphical illustration shows the data filtering steps for single and trio/multisample analysis. After quality control (QC), small variants, CNVs and SVs are filtered for rarity and by the mode of inheritance. Optional filters considering, e.g., variant type, variant impact, coding regions, mosaicism and phenotype can be used on a per-sample basis. AD: autosomal-dominant; AR: autosomal-recessive; XL: X-linked; het: heterozygous; hom: homozygous; comp-het: compound-heterozygous; LOH: loss-of-heterozygosity; ROH: runs-of-homozygosity; CHIP: clonal hematopoiesis of indeterminate potential.

**Figure 2 genes-15-00136-f002:**
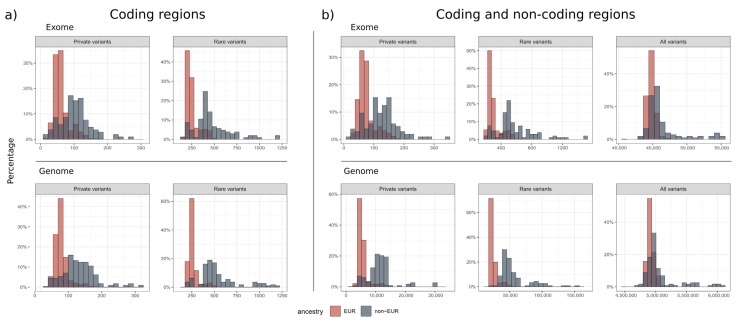
Distribution of rare small variants. The histograms show the percentage distribution of private (MAF = 0%, NGSD count ≤ 1), rare (MAF ≤ 0.1%, NGSD count ≤ 10) and all small variants called in ES and GS, where the European population is shown in red and the non-European population in blue. There were no significant differences in the small variant calls between ES and GS (**a**) in the coding regions. When including the (**b**) non-coding areas, GS generated 82-fold more rare and 75-fold more private variants than ES (Table 3).

**Figure 3 genes-15-00136-f003:**
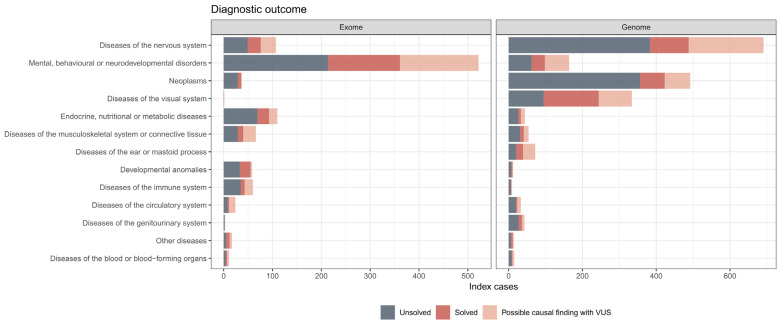
Disease groups and diagnostic outcomes in 2022. Diagnostic ES and GS outcomes are either considered as solved (red), unsolved (blue) or uncertain with possible causal finding with VUS (pink). All solved cases harbored a pathogenic (class 5) or likely pathogenic (class 4) variant evaluated according to the ACMG criteria.

**Table 1 genes-15-00136-t001:** Coverage in the coding regions (OMIM gene exons ± 20 bp).

	Exome	Genome
Sample preparation	SureSelect Human All Exon v7	TruSeq DNA PCR-Free kit
DNA-Sequencer system	NovaSeq 6000 system (Illumina)	NovaSeq 6000 system (Illumina)
Coverage aim	150×	38×
Coverage average	157×	39×
Coverage < 20× with MQ = 0 reads	2.80%	0.23%
Coverage < 20× with MQ ≥ 1 reads only	3.26%	0.63%

MQ: mapping quality.

**Table 2 genes-15-00136-t002:** Benchmarks results for exomes and genomes.

		SNVs			InDels		
		F1	Sensitivity	PPV	F1	Sensitivity	PPV
GiaB high-confidence region, >15× coverage	ES	0.9875	0.9906	0.9844	0.9427	0.9588	0.9273
	GS	0.9955	0.9963	0.9947	0.9858	0.9778	0.9940
Protein-coding exons ± 20 bp, without coverage cutoff	ES	0.9795	0.9720	0.9871	0.9309	0.9277	0.9341
	GS	0.9916	0.9907	0.9924	0.9878	0.9869	0.9888

**Table 3 genes-15-00136-t003:** Variant counts (mean and standard deviation).

	Exome		Genome	
	EUR	Non-EUR	EUR	Non-EUR
Small variants (all)	44,732 ± 595	46,504 ± 2778	4,913,537 ± 56,811	5,080,539 ± 292,286
Small variants—rare (MAF ≤ 0.1%, NGSD count ≤ 10)	302 ± 86	578 ± 253	24,809 ± 5655	58,732 ± 33,912
Small variants—rare, coding (MAF ≤ 0.1%, NGSD count ≤ 10)	252 ± 71	476 ± 203	260 ± 54	575 ± 318
Small variants—private (MAF = 0%, NGSD count ≤ 1)	73 ± 28	126 ± 56	5533 ± 1746	11,989 ± 7523
Small variants—private, coding (MAF = 0%, NGSD count ≤ 1)	61 ± 23	103 ± 46	61 ± 20	120 ± 69
CNVs (all)			10,727 ± 6520	13,668 ± 16,813
CNVs—rare (AF ≤ 5%, good quality, OMIM)			4 ± 6	16 ± 104
SVs (all)			10,907 ± 602	11,108 ± 792
SVs—rare (AF ≤ 1%, good quality, OMIM)			9 ± 4	21 ± 20

## Data Availability

All data presented in this work were processed with our data analysis pipeline megSAP. megSAP is freely available at https://github.com/imgag/megSAP. A list of used open-source tools and databases can be found at https://github.com/imgag/megSAP/blob/master/doc/dna_single_sample.md. Additional data from this study can be provided upon reasonable request.

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
