# Peer review of "Lessons Learned from Translating Genome Sequencing to Clinical Routine: Understanding the Accuracy of a Diagnostic Pipeline"

_genes, 2024, doi:10.3390/genes15010136_

Round 1

Reviewer 1 Report

Comments and Suggestions for Authors

The authors describe the comparative yield of genome sequencing (n=1977) vs exome (n=1016) sequencing in a large cohort from Tübingen.

The article is well written and easy to understand

One interesting characteristic of their study is the difference between the age of patients in the exome  (median = 7) vs genome (median=49 years). This also obviously affected the percentage of cases that were trios (exome, 22% vs genome, 1,8%). This may have negatively influenced the yield in the genome arm as illustrated in numerous studies (e.g. DDD study OR ~5 of reaching a diagnosis in DD/MR with trio design). The authors may wish to comment on this aspect.

The authors may also wish to comment on the yield for triplet expansion between the two groups if tools such as expansion hunter were ran, given that the two arms are not 100% comparable.

Author Response

We would like to thank the reviewer for his/her comments and for the review. We here provide a point-by-point response.

1. This is indeed an interesting question. As suggested by the reviewer we extended the discussion and added more literature to elaborate on the aspects of 1) differences in diagnostic rates for early vs. late onset, and 2) potential methodological advantages of trios (e.g. de novo variants).

2. We thank the reviewer for this suggestion. Two repeat expansions (RE) were detected by ES and 12 RE by GS. However, a direct comparison in terms of yield is not feasible as indicated by the reviewer. Both methods have not been performed on all samples and for a direct comparison, the cases should be best equally selected for phenotypes where repeat disorders are more common (e.g. ataxia or ALS), which we have not done. We used the Expansion Hunter tool to screen 39 different repeat motives/regions (Supplementary Table 5 included). However, we observed that, on average, 16 (mostly varying from 15-17) of these regions showed LowDepth in ES and were therefore not eligible for analysis. This suggests that GS has higher yield/sensitivity for repeat disorders. We added this observation to the discussion.

Reviewer 2 Report

Comments and Suggestions for Authors

Please find my specific comments below.

The main question addressed by the research regards the use of genome sequencing (GS) in diagnostic pathway in rare diseases and comparison the results of two techniques, exome sequencing (ES) and GS. I find the topic original and relevant in the field. It addresses a specific gap in the genetic diagnostics and presents novel results comparing with other studies.

In my opinion the manuscript does not need specific improvements in the field of methodology or controls. The conclusions are consistent with the presented results and address the main purpose of the paper. The list of references is appropriate.

As said before, I would recommend adding a list of specific diagnoses solved by ES/GS, maybe as a Table in the main text or as an additional file. It is of great relevance for clinicians taking care of the patients with rare diseases.

Author Response

We thank the reviewer for this suggestion. As proposed, we added a list of all solved genes in order to illustrate the specific diagnoses that have been solved by ES and GS (Supplementary Table 4).